# Integrated Transcriptome and Proteome Analyses Reveal the Regulatory Role of miR-146a in Human Limbal Epithelium via Notch Signaling

**DOI:** 10.3390/cells9102175

**Published:** 2020-09-26

**Authors:** Adam J. Poe, Mangesh Kulkarni, Aleksandra Leszczynska, Jie Tang, Ruchi Shah, Yasaman Jami-Alahmadi, Jason Wang, Andrei A. Kramerov, James Wohlschlegel, Vasu Punj, Alexander V. Ljubimov, Mehrnoosh Saghizadeh

**Affiliations:** 1Board of Governors Regenerative Medicine Institute, Eye Program, Cedars-Sinai Medical Center, Los Angeles, CA 90048, USA; Adam.Poe@cshs.org (A.J.P.); drmangeshk@gmail.com (M.K.); aleksandra.leszczynska@gmail.com (A.L.); Ruchi.Shah@cshs.org (R.S.); do21.jason.wang@nv.touro.edu (J.W.); Andrei.Kramerov@cshs.org (A.A.K.); ljubimov@cshs.org (A.V.L.); 2Department of Biomedical Sciences, Cedars-Sinai Medical Center, Los Angeles, CA 90048, USA; 3Genomics Core, Cedars-Sinai Medical Center, Los Angeles, CA 90048, USA; JayTang@live.com; 4Department of Biological Chemistry, University of California, Los Angeles, CA 90095, USA; yasaman.jami@gmail.com (Y.J.-A.); jwohl@ucla.edu (J.W.); 5Department of Medicine, University of Southern California, Los Angeles, CA 90089, USA; vasupunj@yahoo.com; 6David Geffen School of Medicine at UCLA, Los Angeles, CA 90095, USA

**Keywords:** cornea, miRNA, miR-146a, Notch, Numb, limbal stem cells, proteomics, transcriptomic, RNA-seq

## Abstract

MiR-146a is upregulated in the stem cell-enriched limbal region vs. central human cornea and can mediate corneal epithelial wound healing. The aim of this study was to identify miR-146a targets in human primary limbal epithelial cells (LECs) using genomic and proteomic analyses. RNA-seq combined with quantitative proteomics based on multiplexed isobaric tandem mass tag labeling was performed in LECs transfected with miR-146a mimic vs. mimic control. Western blot and immunostaining were used to confirm the expression of some targeted genes/proteins. A total of 251 differentially expressed mRNAs and 163 proteins were identified. We found that miR-146a regulates the expression of multiple genes in different pathways, such as the Notch system. In LECs and organ-cultured corneas, miR-146a increased Notch-1 expression possibly by downregulating its inhibitor Numb, but decreased Notch-2. Integrated transcriptome and proteome analyses revealed the regulatory role of miR-146a in several other processes, including anchoring junctions, TNF-α, Hedgehog signaling, adherens junctions, TGF-β, mTORC2, and epidermal growth factor receptor (EGFR) signaling, which mediate wound healing, inflammation, and stem cell maintenance and differentiation. Our results provide insights into the regulatory network of miR-146a and its role in fine-tuning of Notch-1 and Notch-2 expressions in limbal epithelium, which could be a balancing factor in stem cell maintenance and differentiation.

## 1. Introduction

The corneal epithelium is continuously renewed by expansion of limbal epithelial stem cells (LESCs) residing at the corneoscleral junction, the limbus [1,2]. As part of regular corneal epithelial homeostasis, LESCs give rise to transient amplifying (TA) cells, which migrate towards the central cornea and differentiate into mature epithelial cells [3]. Proper renewal and healing of the corneal epithelium is required to maintain corneal transparency and protection of the eye from the external environment. Pathological conditions, including diabetes mellitus (DM), can disrupt epithelial homeostasis, resulting in abnormal epithelial self-renewal and dysfunctional wound healing. Previously, we found that the diabetic corneal epithelium has significantly decreased expression of several putative LESC markers, resulting in impaired corneal epithelial wound healing [4]. Failure of LESC maintenance and subsequent epithelial renewal can result in decrease in corneal transparency, leading to vision loss.

Numerous studies have documented an important role of microRNAs in regulating multiple cellular functions, including differentiation, proliferation, and migration [4,5,6]. These short, noncoding RNAs regulate gene expression at the post-transcriptional level by binding the 3′-UTR of messenger RNA to target the transcript for degradation and/or blocking ribosomal translation [7,8]. We have reported differences in miRNA expression in the limbus and central cornea, along with alterations in diabetic corneas [9]. miR-146a was found to be enriched in the limbus and had increased overall expression in diabetic vs. normal corneas. In addition, changes in miR-146a expression altered corneal wound healing [9,10]. Overexpression of miR-146a resulted in delayed wound closure in human limbal epithelial cells (LECs) in vitro and in ex vivo organ-cultured corneas by direct targeting of epidermal growth factor receptor (EGFR) expression. Inhibition of miR-146a greatly enhanced cell migration and wound healing in human organ-cultured diabetic corneas [10]. Interestingly, miR-146a overexpression also resulted in an increase in putative LESC marker, keratin 15 (K15) [10]. It has also been shown that miR-146a plays a role in regulating hematopoietic stem cell differentiation and survival [11] and the NF-κB inflammatory responses [12]. In the present study, to better understand the regulatory role of miR-146a in the limbal epithelium, we have investigated the potential targets of miR-146a in LECs using RNA-seq transcriptomics combined with quantitative proteomics analysis based on multiplexed isobaric tandem mass tag (TMT) labeling. Since miRNAs suppress the expression of the target genes, miR-146a overexpression was expected to show negative correlation with putative miRNA target gene/protein pairs. We found that miR-146a regulates the expression of a large number of genes in different pathways at the mRNA and protein levels such as Notch signaling, a developmentally conserved signaling pathway that functions in limbal epithelial cells [13,14,15,16]. In the present study, we focused on the Notch signaling system, since both our proteomics and transcriptomics data suggested Notch signaling as a common pathway significantly altered by miR-146a in the human limbal epithelium. Additionally, gene set enrichment analysis (GSEA) of transcriptome data and the overall enrichment score revealed that the Notch pathway was one of the most significantly altered and enriched pathways. Furthermore, Notch signaling is a distinct pathway of interest in limbal epithelial stem cell maintenance, which is known to regulate corneal epithelial homeostasis, and controls cell-fate specification events during wound healing [17,18,19,20]. The functions of miR-146a target genes were further analyzed by Ingenuity pathway analysis. Our study documented the large impact of miR-146a on gene expression at the mRNA and protein levels, providing insights into the regulatory role of miR-146a in limbal epithelial progenitor cells.

## 2. Materials and Methods

### 2.1. Human Corneas

Human cadaver corneas (Table 1) were received from the National Disease Research Interchange (NDRI, Philadelphia, PA, USA) in Optisol storage medium (Chiron Vision, Claremont, CA, USA). The study was conducted in accordance with the Declaration of Helsinki. The NDRI uses a human tissue collection protocol approved by a managerial committee and subject to National Institutes of Health oversight. Samples were received within 24 h of procurement. This work was covered by IRB protocol Pro00019393 from Cedars-Sinai Medical Center. 

### 2.2. Primary Limbal Epithelial Cell Isolation and Cell Culture Maintenance

Primary limbal epithelial cells (LECs) were isolated from human corneal limbus as previously described and characterized as stem-cell-enriched cell population [9,21,22,23]. Briefly, epithelial cells were removed from corneoscleral rims following Dispase/Trypsin digestion. Cells were placed in plates coated with a mixture of fibronectin, collagen IV, and laminin-521 [10,24]. Cells were cultured in EpiLife medium containing Human Keratinocyte Growth supplement (HKGS), N-2 supplement, B27 supplement, and 1X antibiotic/antimycotic mixture with added 10 ng/mL EGF (Thermo Fisher Scientific, Waltham, MA, USA) [10]. Telomerase-immortalized human corneal epithelial cells (HCECs) obtained from Dr. S. Dan Dimitrijevich [9] were grown on type IV collagen-coated plates in EpiLife serum free medium (Thermo Fisher Scientific) with HKGS at 37 °C, 5% CO_2_.

### 2.3. miRNA Transfection of Primary LECs and HCECs

Human primary LECs were transfected with 50 nM hsa-miR-146a-5p pre-miR miRNA precursor or anti-miR miRNA inhibitor, or their respective negative controls using Lipofectamine RNAiMAX transfection agent (all from Thermo Fisher Scientific) following the manufacturer’s instructions. Cells were harvested 3 days post-transfection for protein and RNA analysis or processed for immunostaining. Corneal organ cultures [10] were maintained in Dulbecco’s Modified Eagle’s Medium with 1X insulin-transferrin-selenite (Sigma-Aldrich, St Louis, MO, USA) and 1X antibiotic/antimycotic mix (Thermo Fisher Scientific).

### 2.4. Total RNA Isolation

Total RNA was extracted using the PureLink RNA mini kit after Trizol lysis (Thermo Fisher Scientific) following the manufacturer’s instructions. An on-column DNase digestion was performed using the PureLink DNase Set (Thermo Fisher Scientific). Samples were eluted in RNase-free water. RNA concentration and quality were assessed using a NanoDrop 8000 spectrophotometer (Thermo Fisher Scientific), Qubit 2.0 Fluorometer (Thermo Fisher Scientific), and Agilent 2100 bioanalyzer (Agilent Technologies, Santa Clara, CA, USA).

### 2.5. Quantitative Real-Time RT-PCR (qRT-PCR)

RT-PCR was performed as described previously [9]. Briefly, 10 ng of total RNA was reverse transcribed (RT) using a Taqman microRNA RT kit (Thermo Fisher Scientific) and specific RT primers. Q-PCR was carried out using Taqman 2X universal PCR master mix (no AmpErase UNG) and 20X MicroRNA Assays (Thermo Fisher Scientific) with specific primer for miR-146a, which is designed to detect and quantify mature miRNAs in real time using a 7300 PCR System (Thermo Fisher Scientific). Each sample was run in triplicate. Signals were normalized to U6 housekeeping miRNA run simultaneously. A comparative threshold cycle (Ct) method (ΔΔCt) was used to calculate relative miRNA expression between treated and corresponding control-transfected samples.

### 2.6. Library Preparation and RNA Sequencing

Library construction was performed using the Illumina TruSeq Stranded mRNA library preparation kit (Illumina, San Diego, CA, USA). Briefly, total RNA from eight individual human corneas was assessed for concentration using a Qubit fluorometer and for quality using the 2100 Bioanalyzer. Up to one µg of total RNA per sample was used for poly-A mRNA selection. cDNA was synthesized from enriched and fragmented RNA using reverse transcriptase (Thermo Fisher Scientific) and random primers. The cDNA was further converted into double-stranded DNA (dsDNA), and the resulting dsDNA was enriched with PCR for library preparation. The PCR-amplified library was purified using Agencourt AMPure XP beads (Beckman Coulter, Brea, CA, USA). The concentration of the amplified library was measured with a Qubit fluorometer and an aliquot of the library was resolved on a Bioanalyzer. Sample libraries were multiplexed and sequenced on a NextSeq 500 platform (Illumina) using 75 bp single-end sequencing. On average, about 25 million reads were generated from each sample.

### 2.7. Next-Generation Sequencing Data Analysis

Raw reads obtained from RNA-seq were aligned to the transcriptome with STAR (version 2.5.0) [25]/RSEM (version 1.2.25) [26] with default parameters, using a custom human GRCh38 transcriptome containing all genes coding for protein and long non-coding RNA based on human the GENCODE version 23 annotation. Reads were quantified as the number of reads across exons. Differentially expressed genes were identified by combining two approaches that use different algorithms. A gene was called differentially expressed if it passed the false discovery rate (FDR) with adjusted *p* < 0.05 in DEseq2 [27] and employing the null model hypothesis [28], as has been detailed previously [29]. To investigate potential functional enrichment in differentially expressed genes of various biological pathways by RNA-seq, a ranked *p* value was computed for each pathway from the Fisher exact test based on the binomial distribution and independence for probability of any gene belonging to any enriched set [30]. Unless specified, hierarchical clustering, principal component analysis, and statistical analysis were performed in R v3.0 (http://www.r-project.org). 

Ingenuity pathway analysis (IPA) was used to determine the molecular activities of the differentially expressed genes. Upstream regulator analysis was used to predict the activation or inhibition of upstream regulators [31]. The *p* value of the enrichment score was used to evaluate the significance of the overlap between observed and predicted gene sets, whereas the activation Z score was used to assess the match between observed and predicted patterns of activation or inhibition (Z ± 2). 

To identify pathways that are most significantly altered in RNA-seq, a statistically significant enrichment score (ES) was calculated from a set of ranked genes that are enriched within a single pathway. Then, an overall combined enrichment score and odd ratio were calculated to rank various pathways.

Nonparametric Gene Set Enrichment Analysis (GSEA) method was used to study the relative importance of Notch pathway in our transcriptome data. GSEA-ranked genes were according to their relative expression in 146aM- vs. mimic control (MC)-transfected LECs. Using GSEA, we compared this ranked list of genes to a large collection of pathway gene sets derived from a molecular signature database repository (MSig) from Broad Institute and assigned an enrichment score. The enrichment statistic is the maximum deviation of the running enrichment score from zero. The gene sets that significantly out-performed the random-class permutations were considered significant, as detailed in our previous study [32]. A significance threshold was set at a nominal *p*-value of 0.05. Furthermore, a subset of genes known as core genes, that contribute to the enrichment result, were identified, as detailed previously [33]. 

### 2.8. Protein Extraction and Liquid Chromatography-Mass Spectrometry (LC-MS/MS) Analysis

Primary LECs isolated from individual human corneas (n = 4) were transfected with miR-146a mimic (M) and its respective control (MC). Three days after transfection, the cells were collected, and cell lysates were prepared for proteomics analysis. Briefly, the protein lysates were reduced, alkylated, and digested after addition of lys-C and trypsin proteinases, as previously described [34,35]. Peptides were then labeled using 10-plex TMT isobaric tags [36]. Labeled samples were mixed, desalted, and then loaded onto a fused silica capillary packed in-house with bulk C18 reversed phase resin (length, 25 cm; inner diameter, 75 µM; particle size, 1.9 µm; pore size, 100 Å; Dr. Maisch GmbH, Germany). 

The peptides were separated using a Dionex Ultimate 3000 UHPLC system (Thermo Fisher Scientific). The 140-min water–acetonitrile gradient was delivered at a flow rate of 300 nL/min (Buffer A: water with 3% Dimethyl Sulfoxide (DMSO) and 0.1% formic acid, Buffer B: acetonitrile with 3% DMSO and 0.1% formic acid). Peptides were ionized, introduced into the Orbitrap Fusion Lumos mass spectrometer (Thermo Fisher Scientific), and analyzed by tandem mass spectrometry.

### 2.9. Proteomics Data Analysis

The MS/MS proteomics data were acquired using the synchronous precursor-selection-based MS3 method to minimize reporter ion interference. Database searching and the extraction of TMT reporter ion intensity was performed using the Proteome Discoverer software. Comparison of TMT data across samples was performed using MS Stats to identify proteins that were differentially expressed between the two conditions [37], and a *p*-value was calculated between two groups in a t-test. In addition, expression data were analyzed in conjunction with TargetScan to increase the likelihood of finding direct miRNA targets.

### 2.10. Western Blot Analysis

Western blot was performed as described previously [9]. Briefly, equal amounts of cell lysates were subjected to SDS-PAGE using 8% to 16% gradient Tris-glycine SDS polyacrylamide gels (Thermo Fisher Scientific). Proteins were transferred to nitrocellulose membranes and blocked using Blotting Grade Blocker (Bio-Rad, Hercules, CA, USA) before being probed with primary antibodies (Table 2). IRDye 800 CW or 680 RD goat anti-mouse or anti-rabbit secondary antibodies (LI-COR Biosciences, Lincoln, NE, USA) were used. Blots were imaged in an Odyssey CLX imaging system (LI-COR Biosciences).

### 2.11. Immunostaining

Cultured primary LECs or 5 µm thick transverse corneal cryostat sections were fixed in 4% paraformaldehyde (PFA) and 1% paraformaldehyde, respectively. Immunostaining of fixed LECs and corneal sections was performed after permeabilization in 0.2% Triton X-100 for 10 min at room temperature, as previously described [10]. Slides were mounted with ProLong Gold Antifade Mountant with DAPI (Thermo Fisher Scientific). For each marker, the same exposure time was used when taking pictures. Negative controls without a primary antibody were included in each experiment.

### 2.12. Statistical Analysis

Western blot densitometry was analyzed using Student’s *t*-test for two groups, with *p* < 0.05 considered significant.

## 3. Results

### 3.1. Transcriptomics Analysis of miR-146a-Transfected LECs

To identify miR-146a target genes, primary human LECs from individual donors (*n* = 8) were transfected with miR-146a mimic and its corresponding negative control. Total RNA was isolated 72 h post-transfection, and qRT-PCR analysis confirmed significant upregulation of miR-146a in mimic-transfected LECs (Appendix A). mRNA was sequenced and analyzed as described in Materials and Methods. Principal component analysis (PCA) showed segregation of miR-146a mimic (M) and mimic control (MC)-transfected samples into two distinct principal groups (Figure 1A). A set of 251 genes (Appendix A) was identified as differentially expressed in miR-146aM vs. MC with FDR-adjusted *p* < 0.05 and fold change (FC) of ±1.5. A two-way hierarchical clustering using Euclidean distance and average linkage (Figure 1B) showed a clear distinction of 63 mRNAs upregulated and 188 mRNAs downregulated in miR-146aM- vs. MC- (*p* < 0.05 and FC ± 1.5) transfected LECs (Appendix A).

Overexpression of miR-146a in primary LECs significantly decreased mRNA expression levels of known target genes, such as *EGFR*, which we have shown previously in relation to slow wound healing in vitro and in ex vivo organ-cultured diabetic corneas [9]. Analysis also showed changes in expression of the Notch signaling pathway members. There was a decrease in expression of *NUMB*, coding for a Notch-1 inhibitor, in cells treated with miR-146a mimic; interestingly, there was also a concomitant increase in *NOTCH-1*, whereas *NOTCH-2* mRNA expression level was decreased (Appendix A). We also found a decrease in the mRNA expression level of known regulators of the *NF-κB* inflammatory pathway, *IRAK1, TRAF6*, and chemokines (*CXCL1, CXCL2, CXCL5*, and *CXCL8*). There was also an increase of putative stem cell marker *KRT15* in miR-146aM compared to MC, which was consistent with our previous finding [10]. 

Furthermore, we used the gene set enrichment analysis (GSEA) method to prioritize candidate genes in the pathway. GSEA showed that genes corresponding to the Notch pathway were significantly enriched (normalized enrichment score (NES) = 2.12; *p* < 0.05) in our transcriptome dataset (Figure 1C,D). Normalized read counts corresponding to miR-146aM and MC were used to rank the enrichment of genes (green line) in the molecular signature database of the Broad Institute using the GSEA algorithm. Among the 18 ranked core enriched genes based on their enrichment scores, *NOTCH-1* is the most significantly enriched gene in GSEA (Figure 1D). The core genes are a subset of genes that drive the enrichment of specific pathways.

### 3.2. Proteomics Analysis of miR-146a-Transfected LECs

Primary human LECs (*n* = 4) transfected with miR-146aM vs. MC were analyzed for differential expression of protein targets by LC-MS/MS. A total of 163 proteins were identified as differentially expressed with the FDR-adjusted *p* < 0.05 and fold change (FC) of ±1.5 (Appendix A). There were 50 proteins upregulated and 113 proteins downregulated with FDR-adjusted *p* < 0.05 in miR-146aM- vs. MC-transfected LECs, as shown and verified by boxplot analysis (Figure 2A). A heat map of 163 differentially expressed proteomic targets in miR-146a-transfected LECs divided two groups into distinct clusters (Figure 2B).

Known inflammatory targets, such as Traf6, IL-1α, and IL-1β, as well as Notch-1 inhibitor, Numb, were downregulated, whereas Notch-1 and putative stem cell marker K15 were upregulated in miR-146aM-transfected cells, in agreement with our RNA-seq data. Proteomic analysis was consistent with our RNA-seq data and our previously documented decrease in EGFR protein level in miR-146aM-transfected LECs [10]. Integration of proteomic and genomic features showed 70 overlapping differentially expressed targets (Appendix A and Appendix A), which were present in both RNA-seq and proteomics (FDR *p* < 0.05, FC ± 1.2). 

### 3.3. Functional Analysis of Differentially Expressed mRNAs/Proteins

Ingenuity pathway analyses of proteomics and transcriptomics data found significant changes in multiple signaling pathways. Pathway analysis of differentially expressed mRNAs in miR-146aM vs. MC showed significant differences in inflammatory signaling pathways such as IL-6 and IL-8, immune response signaling pathways such as endothelin-1, iNOS, and PI3K, and in the Notch signaling pathway (Figure 3A). Additionally, Notch signaling was identified as one of the most significantly altered pathways (FDR *p* < 0.05), with a combined enrichment score of 85 and Odd ratio of 12 (Appendix A).

Analysis of proteomic data found significant changes in Toll-like Receptor (TLR) signaling, Stat3, senescence, eIF2, integrin, and Notch signaling (Figure 3B). Integration of transcriptomics and proteomics data revealed similar significantly enriched common functional pathways, including anchoring and adherens junctions, NF-κB, TNF-α, Hedgehog signaling, TGF-β, mTORC2, EGFR, and Notch signaling (Figure 3C). Although all these pathways play important roles in corneal epithelial homeostasis and as potential targets of miR-146a, interestingly, the Notch signaling pathway was identified by both our proteomics and transcriptomics as a potential target of miR-146a, which also seems to play an important role in limbal progenitor cell homeostasis.

Examination of upstream regulators in our RNA-seq pathway data showed inhibition of signaling molecules, such as IL-1α/β, ERK, p38 MAPK, NF-κB, IL-6, and Notch signaling regulators such as EHF, 26s proteasome, and JAG2, as well as a differentially expressed regulator, TNF (Figure 4A). Proteomic upstream regulator analysis indicated inhibition of signaling molecules, such as IL-1α/β, EGFR (differentially expressed regulators), ERK, and Notch signaling regulators TNF and TCF4, as well as activation of PI3K and p63 (Figure 4B).

### 3.4. miR-146a Alters Notch Signaling in Human Primary LECs, HCECs, and Organ-Cultured Corneas

Both our transcriptomics and proteomics analyses revealed Notch signaling molecules as potential targets of miR-146a. This is a distinct pathway, which is known to regulate corneal epithelial homeostasis. Changes in Notch proteins in miR146aM-transfected cells were further explored and confirmed by western analysis and immunostaining. Upregulation and downregulation of miR-146a were confirmed by qRT-PCR in mimic- and inhibitor-transfected cells, respectively (Appendix A). Overexpression of miR-146aM in primary LECs significantly decreased protein expression of its putative target, Numb (Notch-1 inhibitor), relative to control-transfected cells, and miR-146a inhibitor had an opposite effect (Figure 5A). Accordingly, there was a concomitant significant increase in Notch-1 expression (Figure 5A). Notch-2, another putative direct target of miR-146a, was also downregulated significantly (Figure 5B) in miR-146aM- vs. MC-transfected LECs. Silencing of miR-146a using its inhibitor led to some decrease of Notch-1 and modest increase of Notch-2 protein expression, neither of which reached significance (Figure 5A,B). Furthermore, western blot analysis of Hes1, a downstream intermediate of the Notch signaling pathway, showed no significant changes in miR-*146aM-* vs. MC-transfected LECs (Figure 5B). Immunostaining of human primary LECs treated with miR-146aM showed a similar increase in Notch-1 and decrease in Numb and Notch-2 compared to cells transfected with MC (Figure 5C). However, Hes1 showed no significant change in expression level in mimic- vs. MC-transfected cells (Figure 5C) in line with the western blot.

Both HCECs and human organ-cultured corneas treated with miR-146a compared to the cells and the fellow corneas transfected with scrambled sequences, respectively, showed a similar increase in Notch-1 and decrease in Numb protein levels, whereas the inhibitor had opposite effects (Figure 6A). Notch-2 (300 kD), along with its active form, Notch intracellular domain (NICD,110 kD), was significantly downregulated in miR-146aM-transfected HCECs, whereas miR-146a inhibitor caused a subtle increase of both 300 kD Notch-1 and NICD levels according to the western blot (Figure 6B). Immunostaining of human organ-cultured corneas treated with miR-146aM showed a similar increase in Notch-1 and decrease in Numb and Notch-2 in limbal epithelium compared to fellow corneas transfected with MC (Figure 6C). Noteworthily, Notch-1 was found predominantly in the basal and immediate suprabasal layer in limbus (Figure 6C, arrowheads), whereas, Notch-2 was detected in the basal, suprabasal, and, to some extent, the superficial layer in limbus (Figure 6C, arrowheads) by immunohistochemistry. The downstream Notch signaling target, Hey1, showed an increased expression level in mimic-transfected corneas in the limbus (Figure 6C), with opposite expression in the central cornea (Appendix A); however, there was no significant change in Hes1 expression level in either the limbal or central cornea (Figure 6C and Appendix A) compared to the MC-transfected fellow corneas.

### 3.5. miR-146a in Limbal Epithelial Stem Cell Maintenance

Previously, we observed increased expression of putative LESC marker K15 in miR-146aM- vs. MC-transfected primary LECs by immunostaining [10]. Both our transcriptomic and proteomic analyses of normal LECs treated with miR-146aM confirmed this finding [10]. The effects of miR-146a mimic and inhibitor on K15 expression were also examined by immunostaining of human organ-cultured corneas and western blot of primary LECs (Figure 7).

Overexpression of miR-146a in human organ-cultured corneas increased, whereas its inhibition decreased K15 expression level compared to their corresponding control fellow corneas according to immunostaining (Figure 7A). In primary LECs, miR-146aM treatment also led to an increase in K15 expression vs. MC, whereas silencing of miR-146a with its inhibitor led to modestly decreased K15 expression according to the western blot (Figure 7B). Furthermore, we set out to investigate the effect of Notch inhibitor (γ-secretase inhibitor, DAPT) on expression of K15. Our results showed that DAPT treatment reverted the effect of miR-146a in upregulation of K15 expression in miR-146aM-transfected cells (Figure 8), suggesting that the effect of miR-146a on limbal stem cells depends on the activation of Notch-1. K15 significantly decreased in miR-146aM (M)- and mimic control (MC)-transfected cells compared to control DMSO-treated cells. Significant downregulation of Notch-1 in DAPT-treated LECs was confirmed by western blot; however, downregulation of Notch-2 did not reach significance (Figure 8).

## 4. Discussion

The use of quantitative proteomic along with transcriptomic strategies has emerged as a key technique for a comprehensive identification of miRNA targets. It allows direct determination of proteins with altered levels of expression because of translational suppression without mRNA degradation [38,39]. Therefore, by performing a combined transcriptomic and proteomic profiling, we provide the first comprehensive molecular insight into the miR-146a regulatory roles in the corneal epithelium. Previously, we have reported upregulation of a number of miRNAs, including miR-146a, in the stem-cell-enriched limbal region vs. central cornea and in diabetic vs. normal corneas [9]. LESC health is a key factor in corneal epithelial homeostasis, regeneration, and wound healing, which are required to maintain corneal transparency and visual acuity. Therefore, understanding the regulatory roles of miRNAs by delineation of their targets, mRNAs and their protein products, is a key to understanding the mechanisms that underlie LESC function.

miR-146a is an important regulator of many cellular functions and targets different genes in different cell types [40,41,42]. It plays a major role in several diseases, such as diabetes [40], cancer [41], and Graves ophthalmopathy [42], and its association with different pathways suggests its involvement in cell migration [9,10,43], invasion [44], differentiation [45], and proliferation [44]. Our previous studies also revealed that miR-146a plays a regulatory role in corneal wound healing and limbal epithelial stem cell maintenance in normal and diabetic corneas [9,10]. 

In the present study, as a further step to investigate the regulatory role of miR-146a, we applied combined transcriptome and proteome analyses to establish miR-146a target mRNA/proteins in corneal epithelium. This led to the identification of a number of miR-146a target genes in LECs, and confirmed our previous findings related to EGFR [9,10]. Integrated RNA-seq and proteomic pathway analyses indicated significant changes in the EGFR pathway, which has a pronounced effect on corneal wound healing [9,10]. Some miR-146a targets have not been documented previously in the corneal epithelium, such as *NUMB, NOTCH-1, NOTCH-2, TRAF6*, *IL-1α,* and *IL-1β* (Appendix A). LEC treatment with miR-146a mimic resulted in significant changes in inflammation-related pathways, including inhibition of upstream regulators *NF-κB*, *TNF, IL-1α,* and *IL-1β* (Figure 3 and Figure 4). miR-146a dysregulation has been reported in several human inflammatory diseases, such as cystic fibrosis [46], hepatic inflammation [47], and diabetic retinopathy [48]. Further investigation is needed to determine how these changes may affect both acute inflammatory challenge as well as chronic disease-related inflammation in the corneal epithelium. In addition, integrated RNA-seq and proteomic pathway analyses revealed changes in Hedgehog signaling and TGF-β (Figure 3C). Study of Hedgehog in mice indicates that it may play a role in both corneal epithelial regeneration and limbal epithelial maintenance [49,50,51]. TGF-β regulates several essential cellular processes in the cornea, such as local inflammatory responses and wound healing in ocular surface epithelial cells [52]. In addition, several studies showed that inhibition of TGF-β signaling enriches and expands p63+ corneal epithelial progenitor cells [53]; both TGF-β1 or TGF-β2 significantly inhibited primary human LEC proliferation [54]. 

Furthermore, IPA analysis of our transcriptome data showed overall significant enrichment of cell growth, cell cycle, differentiation, survival, and inflammatory response genes (Figure 3). Interestingly, among the various pathways enriched in this analysis, Notch signaling was significantly altered in both proteomic and transcriptomic data. Furthermore, a GSEA analysis of transcriptome data showed significant enrichment (*p* < 0.05), with an enrichment score of 2.12 for Notch-1 pathway (Figure 1C,D). Additionally, among all the pathways enriched in transcriptomic data, Notch signaling was identified as one of the most significantly altered pathways (Appendix A); its role in human limbal epithelium is not well understood. Therefore, we focused on Notch signaling pathway.

Notch signaling has been a specific pathway of interest in limbal epithelial stem cell maintenance [55]. There are four mammalian Notch proteins that are large transmembrane receptors. Upon binding a ligand, the intracellular domain (NICD) is cleaved and translocated to the nucleus, modulating gene expression. It is well established that Notch signaling plays a pivotal role in cellular communication, regulating stem cell maintenance, proliferation, and differentiation during embryonic development, as well as tissue homeostasis and stem cell self-renewal in multiple adult organ systems [55,56,57]. The role of Notch in controlling stem cell population and differentiation decisions is considered to be cell context specific [55,58,59]. There have been some conflicting reports regarding the regulatory role of Notch signaling in the corneal epithelium. Inhibition of Notch signaling by small molecule inhibitors decreased levels of both differentiation and proliferation markers in rat LECs [60] and in human limbal stem cells [61]. In contrast, Ma et al. demonstrated that inhibition of Notch signaling resulted in decreased proliferation but increased differentiation in vitro, with no effect in a three-dimensional, stratified corneal epithelium equivalent, whereas Notch activation resulted in decreased differentiation [14]. Additionally, investigation of Notch expression post-wounding found an inverse correlation between Notch signaling and epithelial proliferation. Subsequent restoration of Notch signaling coincided with cellular differentiation [16].

Notch signaling has been found to be directly affected by miR-146a in many tissues [62,63,64]. Our transcriptomics and proteomics data confirmed by western blot analysis, along with in silico analysis of miR-146a-predicted targets with the presence of miR-146a binding site in Notch-2- and Numb 3′-UTRs [45], suggest that both Numb and Notch-2 are novel targets of miR-146a in human primary LECs and organ-cultured corneas (Appendix A, Figure 5 and Figure 6). Overexpression of miR-146a induced repression of Numb and an increase in Notch-1 expression level, which is in agreement with earlier verified miR-146a direct target studies [63,64,65]. This treatment decreased Notch-2 expression level, which is also in agreement with previous data [41,42]. 

Western blot analysis and immunostaining of the Notch signaling target, Hes1, showed no significant changes in miR-146aM- vs. MC-transfected primary LECs and organ-cultured corneas (Figure 6C). However, overexpression of miR-146a in human organ-cultured corneas showed an increase in another Notch signaling target, Hey1, in limbal epithelium (Figure 6C) with an opposite expression pattern, that is, downregulation in the central cornea compared to the control fellow cornea according to immunostaining (Appendix A). Therefore, the spatial upregulation of Hey1 only in the limbal region may suggest its regulation as a downstream target of Notch-1, which is consistent with previous data [66]. However, the lack of significant changes in Hes1 expression level may suggest its regulation by Notch-2 [67] and probably, to some extent, by Notch-1, which explains the lack of significant changes in its expression due to opposing expression patterns of Notch-1 and Notch-2 in the limbal epithelium. In addition, other downstream targets of Notch signaling, such as HeyL, Hes6, P21 (CDKN1A), and/or Cyclin D3 (CCND3), may be involved in activation of Notch signaling in limbal epithelium homeostasis, which needs further investigation.

Interestingly, both miR-146a and Notch-1 were found to be more expressed in the limbal basal and early suprabasal layers where LESC and early TA cells reside [10,13,15,68]. In addition, miR-146a overexpression led to an increased expression of limbal epithelial stem cell marker K15 both in vitro and in organ-cultured corneas, which was reverted by mainly inhibiting Notch-1 signaling. This strongly suggests that the effect of miR-146a on stem cells depends on Notch signaling (Figure 8). Furthermore, our integrated pathway analysis (Figure 3C) indicates significant changes in anchoring and adherens junctions in miR-146aM- vs. MC-transfected LECs. Taken together, these data suggest the role of miR-146a in the regulation of Notch-1 in LESC maintenance and/or asymmetrical cell division leading stem cells to differentiate to early TA cells, which is consistent with previous reports on the regulatory roles of Notch-1 and Numb [69,70]. Downregulation of Notch-2, which has been implicated in differentiation [41], may suggest that there are differential expressions and regulatory roles of Notch family members at different stages of limbal cells undergoing differentiation, from LESCs to terminally differentiated cells. As the committed early TA cells migrate laterally and upward, where miR-146a expression declines, Notch-1 expression decreases, which is in agreement with the study, demonstrating that loss of Notch-1 disrupts the barrier repair with subsequent increase of migratory behavior in the corneal epithelium [71]. Conversely, increased Notch-2 expression coincides with cellular differentiation, which is also in agreement with previous studies [14,16,60,61]. Therefore, miR-146a may orchestrate differential expressions of Notch-1 and Notch-2 in maintaining the population of stem cells and/or their early differentiation. Additionally, it has been shown that Notch-1 and Notch-2 exert opposite regulatory effects on the growth of embryonic brain tumors [72]. These findings may explain the controversial reports regarding the roles of Notch signaling in the corneal epithelium. Overall, the Notch signaling pathway seems to be involved both in maintaining the stem cell population and regulating proper differentiation of progeny cells [41,69,70].

Our study suggests the pivotal regulatory role of miR-146a in corneal epithelial homeostasis in balancing stemness and differentiation of basal cells through modulation of Notch signaling by fine-tuning of Notch-1 and Notch-2 spatiotemporal expressions in the corneal epithelium. This notion is supported by the fact that gene expression silencing by miRNAs could be potentially reversible due to their capability of translational suppression without mRNA degradation, as repressed mRNA could return to functional mode at any given time in response to the requirement of fine-tuning and immediate needs of the cells [73,74,75]. In addition, this may explain the modest changes in protein levels of miR-146a targets according to western blots in our study, which did not reach significance. 

The comparison of miR-146a target gene and protein expression data obtained from transcriptome and proteome profiling indicates good agreement in important signaling pathways, such as EGFR and Notch signaling. However, there are some other cases that are not in concordance, possibly due to one of the miRNA regulatory mechanisms. This represses translation without mRNA degradation [38,39], so that mRNA levels do not necessarily correlate with the levels of protein expression. In addition, there is a discordance between half-lives of mRNAs and proteins, the presence of several isoforms of some proteins, and post-translational modifications of peptides that were not considered in our proteome analysis [76,77,78]. To address the problem of low correlation in proteomics and transcriptomics, a proteogenomics approach [79] can be used. Alternatively, we may extract common functional contexts to identify the consensus biological pathways coordinately regulated by both protein and RNA-seq features (Figure 3C). The existence of such differences necessitates a thorough validation of both methods using additional approaches.

In conclusion, we provided, for the first time, a comprehensive identification of miR-146a targets in the limbal epithelium using integrated global analyses of gene expression and protein profiles that may not be deciphered from individual transcriptomics or proteomics analyses. Overall, all these identified miR-146a-targeted pathways are important for corneal epithelial homeostasis, such as activation of Hedgehog signaling in healing corneal and limbal epithelia [49], TGF-β in corneal wound healing and fibrosis [52], and mTOR, NF-κB, and TNF-α in corneal inflammatory responses and angiogenesis [80], which all need further investigation. We also identified and validated a number of targeted genes and pathways involved in stem cell maintenance and differentiation, such as Notch signaling. We have also uncovered, for the first time, that miR-146a may have an opposite regulatory role in the fine-tuning of Notch-1 and Notch-2 expression in maintaining homeostasis of the corneal epithelium by balancing the stem cell population and differentiation of basal LECs.

## Figures and Tables

**Figure 1 cells-09-02175-f001:**
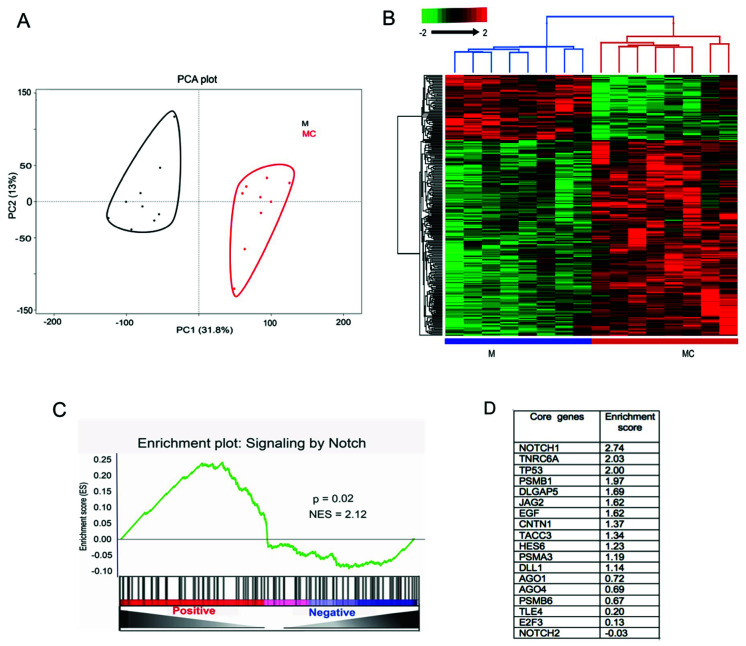
Differentially expressed genes in limbal epithelial cells (LECs) transfected with miR-146a mimic (M) vs. mimic control (MC) identified by RNA-seq. (**A**) Unsupervised principal component analysis (PCA) of the 251 differentially expressed genes shows two distinct clusters expressed in LECs transfected with mimic (black-labeled cluster) or MC (red-labeled cluster). (**B**) Hierarchical clustering of differentially expressed genes, false discovery rate (FDR) *p* < 0.05, using Euclidean distance; average linkage indicates altered expression profiles in M vs. MC samples. Each row represents a single gene, whereas each column represents a sample. The color scale illustrates the relative expression levels of genes across each sample. (**C**) Gene set enrichment analysis; the (GSEA)-enrichment plot shows the enrichment of genes associated with the Notch signaling pathway in our transcriptome dataset. Y-axis plots the enrichment score, and X-axis is the rank of differentially expressed genes in 146aM- vs. MC-transfected LECs; positive indicates upregulation, while negative indicates downregulation in miR-146a-transfected cells. The genes were ranked by increasing expression differences (green curve). (**D**) List of 18 core enriched genes based on ranking of their enrichment scores in the Notch signaling pathway.

**Figure 2 cells-09-02175-f002:**
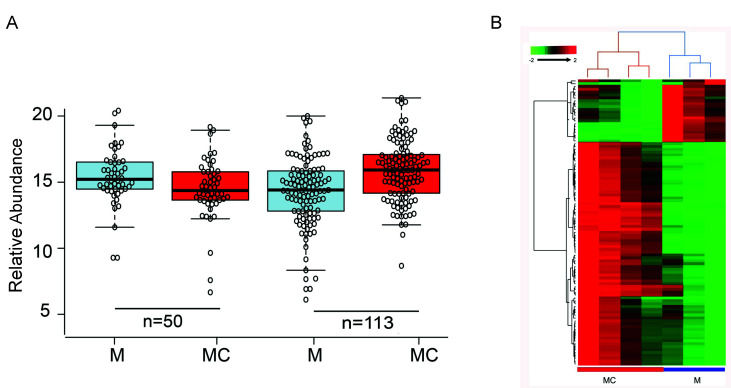
Differentially expressed proteins in miR-146aM- vs. MC-transfected LECs using quantitative proteomics analysis. (**A**) Box whisker plot showing relative abundance of 163 proteins in miR-146aM- vs. MC-transfected LECs. The plot represents the interquartile range (IQR) of the minimum, 25th percentile, median, 75th percentile, and maximum values and shows 50 protein targets that are differentially upregulated, whereas 110 targets are significantly downregulated in M vs. MC at *p* < 0.05. Each circle on the box whisker plot represents expression of a single sample in a respective sample group. (**B**) A two-way hierarchical clustering of 163 differentially expressed protein targets in miR-146a M vs. MC. Euclidean distance and average linkage were used for generating clustering.

**Figure 3 cells-09-02175-f003:**
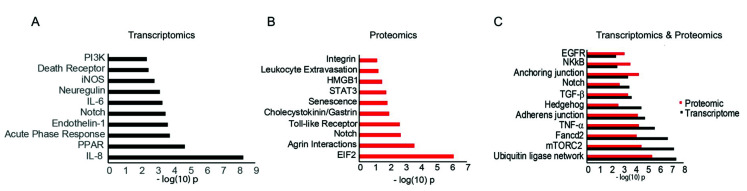
Functional enrichment of pathways in transcriptomics and proteomics of differentially expressed genes/proteins in miR-146aM- vs. MC-transfected LECs. (**A**) Gene set analysis of various signaling pathways enriched in 251 differentially expressed transcriptomic targets in M- vs. MC-transfected samples. (**B**) Enrichment of various pathways in 163 differentially expressed proteomic targets in M- vs. MC-transfected samples. (**C**) Common functional pathways enriched in transcriptomic and proteomic miR-146a targets in M- vs. MC-transfected samples. Each enriched pathway is ranked based on the *p*-value computed from the Fisher exact test based on the binomial distribution and independence for probability of any gene/proteins belonging to any enriched set.

**Figure 4 cells-09-02175-f004:**
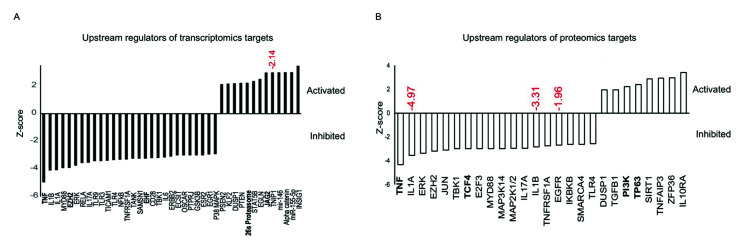
Common upstream regulators altered (activated/inhibited) in transcriptome and proteome. Targets identified in LECs transfected with miR-146aM vs. MC in transcriptome (**A**) and in proteome (**B**). Notch signaling regulators are listed in bold. The activation or inhibition is estimated by a Z-score, which represents the number of standard deviations that a value is away from the mean of a gene value in all the sample groups. Genes that are differentially expressed in our dataset are indicated as fold change in red.

**Figure 5 cells-09-02175-f005:**
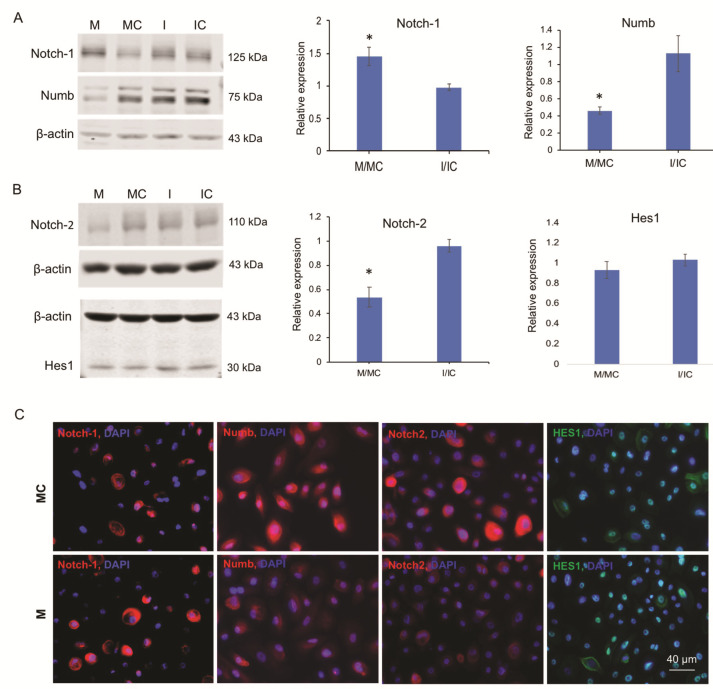
Effect of miR-146a on the expression of Notch signaling molecules in human primary LECs identified by western blot and immunostaining. (**A**,**B**) Total extracted protein from transfected LECs with miR-146a mimic (M) or its inhibitor (I) and their corresponding controls, mimic control (MC) and inhibitor control (IC), respectively, was separated on gradient SDS-PAGE gels, transferred to membrane, and probed with Notch signaling molecule antibodies (Table 2). Antibody to β-actin was used as loading control and for semi-quantitation. (**A**) miR-146a treatment increased, whereas its inhibitor decreased, the protein level of Notch-1. On the contrary, miR-146a treatment decreased, whereas its inhibitor increased Numb expression level in primary LECs compared to their corresponding controls. (**B**) Overexpression of miR-146a in transfected cells decreased, whereas its inhibition increased the protein level of Notch-2 compared to their corresponding controls. However, there was no significant change in Hes1 expression level in treated cells compared to their corresponding controls. The bar graph represents average ± SEM (Standard Error of the Mean) of pooled values (*n* = 4) of densitometric scans. * *p* < 0.05, compared with scrambled control values by paired two-tailed t test. Each bar represents the changes of mimic or inhibitor compared to their corresponding controls, shown as M/MC and inhibitor (I)/inhibitor control (IC). (**C**) Effect of miR-146a on Notch signaling molecule expressions in human primary LECs by immunostaining. Primary human LECs transfected with miR-146a mimic showed increased expression of Notch-1 and decreased expressions of Numb and Notch-2, whereas there was no significant change in Hes1 expression level in miR-146aM-treated cells compared to the mimic control. The same exposure time was used for each set of compared immunostained sections; the pictures are representative of three independent experiments of each transfected primary LEC (*n* = 3).

**Figure 6 cells-09-02175-f006:**
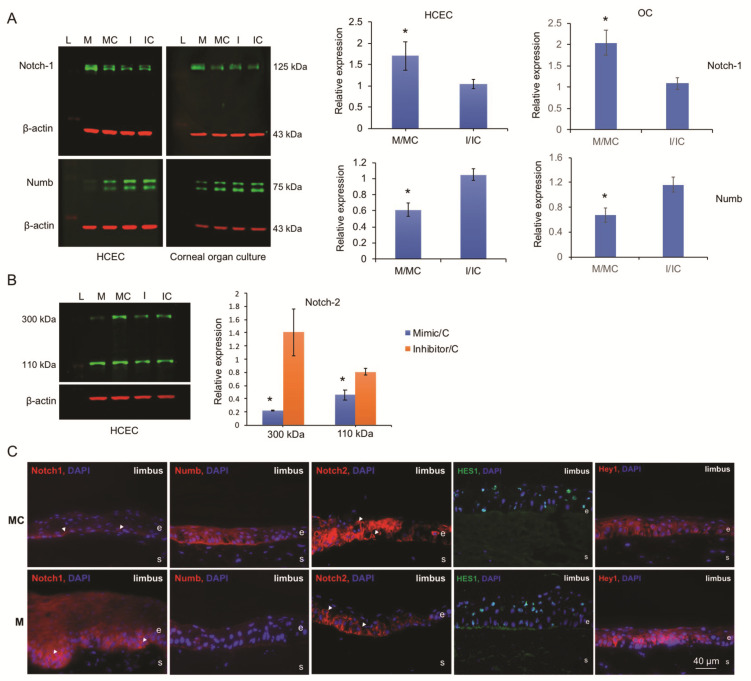
Effect of miR-146a on the expression of Notch signaling molecules in HCECs and human organ-cultured corneas by western blot and immunostaining. (**A**,**B**) Total extracted protein from HCECs and corneas transfected with miR-146a mimic (M) or its inhibitor (I) and their corresponding controls—mimic control (MC) and inhibitor control (IC), respectively—was separated on gradient SDS-PAGE gels, transferred to membrane, and probed with Notch signaling molecule antibodies (Table 2). β-actin was used as loading control and for semi-quantitation. (**A**) miR-146a treatment increased whereas its inhibitor decreased protein level of Notch-1. On the contrary, miR-146a treatment decreased whereas its inhibitor increased Numb expression level. (**B**) Overexpression of miR-146a in transfected HCEC decreased, whereas its inhibition increased protein levels of both 300 kD full-length and about 110 kD in the Notch intracellular domain (NICD) of Notch-2. The bar graph represents average ± SEM of pooled values (*n* = 4) of densitometric scans. * *p* < 0.05, compared with scrambled control values by paired two-tailed t test. Each bar represents the changes of mimic or inhibitor compared to their corresponding controls, shown as M/MC and I/IC. (**C**) Effect of miR-146a on Notch signaling molecule expressions in human organ-cultured corneas by immunostaining. Corneas transfected with miR-146a mimic showed increased expression in the limbus of Notch-1 and Hey1, decreased expression of Numb and Notch-2, and no significant changes of Hes1 expression level compared to their mimic control-transfected fellow corneas. The same exposure time was used for each set of compared immunostained sections; the pictures are representative of three independent experiments of each transfected organ-cultured cornea (*n* = 3). e, epithelium; s, stroma.

**Figure 7 cells-09-02175-f007:**
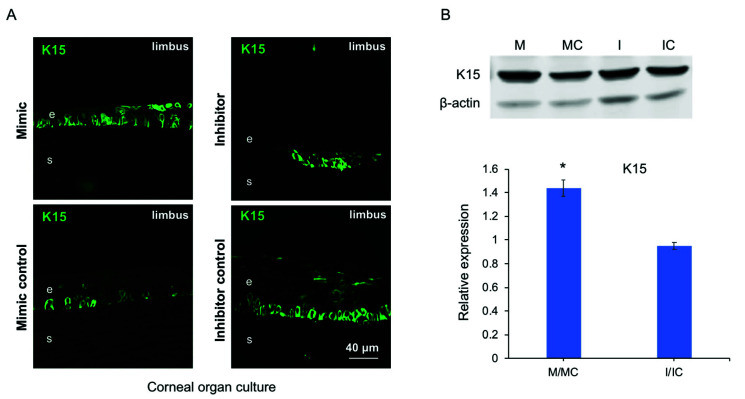
Effect of miR-146a on putative limbal epithelial stem cell (LESC) marker K15 expression in human organ-cultured corneas by immunostaining and in primary LECs by western blot. (**A**) miR-146a transfection increased whereas miR-146a inhibitor decreased staining for K15 in human organ-cultured corneas compared to their corresponding controls. The same exposure time was used for each set of compared immunostained sections, and the assessment was done by more than one observer. The pictures are representative of three independent experiments (*n* = 3). (**B**) miR-146a mimic (M) treatment increased whereas its inhibitor (I) decreased the protein level of K15 in LECs compared to the respective controls. The bar graph represents average ± SEM of pooled values (*n* = 4) of densitometric scans. * *p* < 0.05, compared with scrambled control values by paired two-tailed *t* test. e, epithelium; s, stroma.

**Figure 8 cells-09-02175-f008:**
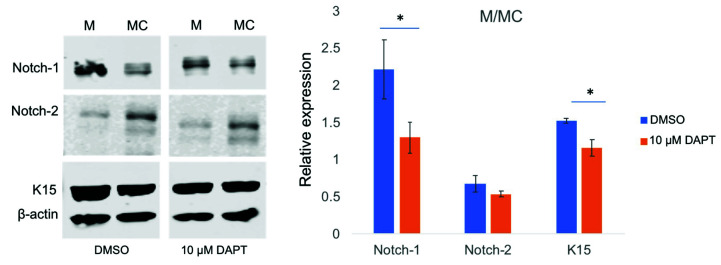
Effect of Notch inhibitor DAPT on miR-146aM-transfected LECs according to the western blot. DAPT treatment (10 μM) reverted the upregulation of K15 expression in miR-146aM (M)- and mimic control (MC)-transfected cells compared to control DMSO-treated cells. Notch-1 and K15 were significantly decreased, whereas downregulation of Notch-2 did not reach significance. The bar graph represents average ± SEM of pooled values (*n* = 3) of densitometric scans. * *p* < 0.05, compared with control values by paired two-tailed *t* test.

**Table 1 cells-09-02175-t001:** Donor characteristics.

Case Number	Age	Gender	Cause of Death	History of Eye Diseases
N16-13	87	M	Ruptured abdominal aortic aneurysm	IOL
N16-14	76	F	Cardiac pulmonary failure	--
N17-04	75	M	Cardiac Arrest	--
N17-06	93	F	Myocardial infarction	IOL
N17-07	73	M	Cardiac arrest	IOL, Cataract
N17-10	80	F	Cerebrovascular/Stroke	IOL, Cataract
N17-11	52	M	Hypertension	IOL
N17-14	51	F	Intracerebral hemorrhage	Cataract
N17-24	75	M	Cardiac arrest	--
N18-19	73	M	Multi-system organ failure	--
N18-23	57	F	Cerebrovascular accident	--
N18-36	74	M	Cardiopulmonary arrest	--
N19-04	71	M	Cardiopulmonary arrest	IOL, Cataract
N19-13	66	M	Cardiopulmonary arrest	Lasik
N19-35	35	M	Potassium chloride overdose	–
N20-03	64	F	Cardiac arrest	–
N20-05	73	M	Shortness of breath	–

IOL, intraocular lens.

**Table 2 cells-09-02175-t002:** Primary antibody list.

Antigen	Antibody	Source	MW (kDa)	Dilution	Application
Notch-1	Rabbit mAb ab52627	Abcam	125	1:500, 1:20, 1:100	WB, IHC, IF
Numb	Rabbit mAb 2761S	Cell Signaling	72, 74	1:500	WB
Notch-2	Goat pAb ab4147 Rabbit mAb 4530S	Abcam Cell Signaling	110	1:20, 1:100 1:500, 1:50, 1:50	IHC, IF WB, IHC, IF
K15	Mouse mAb ab52816	Abcam	45	1:5000, 1:100	WB, IHC
Hey1 Hes1 β-Actin	Rabbit pAb ab22614 Mouse mAb ab119776 Rabbit mAb 11988S Mouse mAb A5441	Abcam Abcam Cell Signaling Sigma	34 30 42	1:50 1:50 1:500 1:2000	IHC IHC, IF WB WB

pAb, polyclonal antibody; mAb, monoclonal antibody; WB, western blot; IHC, immunohistochemistry; IF, immunofluorescence.

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
