# Peer review of "Integrated Transcriptome and Proteome Analyses Reveal the Regulatory Role of miR-146a in Human Limbal Epithelium via Notch Signaling"

_cells, 2020, doi:10.3390/cells9102175_

Round 1
Reviewer 1 Report
The current article by Poe et al provide insights into the regulatory network of miR-146a and its role in fine-tuning of Notch-1 and Notch-2 expressions in limbal epithelium. The authors carried out RNA seq and proteomic analysis on primary LECs post transfection with the miR-146a mimic in order to identify the targets of the miRNA. The study find that Notch signaling is a target by the miR-146a.
Although the study identifies the the downstream target of the miR-146a as the Notch signaling pathway, it doesn't validate and rule out other pathways that may be affected by the miRNA and does not help to improve understanding of the pathway modulation. Functional assays have to be carried out to tweeze out the dependence of the Notch pathway on the miRNA. The following suggestions could help improve the manuscript,
1) Carry out functional assays for downstream targets of the Notch pathway.
2) In Figure 5 and 7, the western blots and the quantification needs to be elaborated as it is confusing. Preferably the controls should be before the treatment group. Try to set the control to 1 and find everything relative to that.
Author Response
Re: Manuscript ID, cells-886180, entitled " Integrated Transcriptome and Proteome Analyses Reveal the Regulatory Role of miR-146a in Human Limbal Epithelium via Notch Signaling". We want to thank the reviewers for a careful evaluation of our submitted paper. We have reviewed all of the concerns and have revised the paper accordingly.
Due to the reviewers’ comments and to make better structure and organization of manuscript, there are some changes in Figures. Figure 6 was changed as Figure 5C; Figure 7 was changed as Figure 6; Figure 8 was changed as Figure 7; a new figure was added as figure 8 (Effect of DAPT (Notch inhibitor) on miR-146aM transfected LECs by western blot). In addition, 3 figures were added as Supplementary Figures, S1, S2, S3. All the original western blots were submitted as Supplementary information.
A new author was added to the list of authors.
Response to Reviewer 1 Comments
Point 1: Carry out functional assays for downstream targets of the Notch pathway.
Response 1: We appreciate pointing this out. The expression levels of Notch downstream targets, Hes1 by western blot (added to Figure 5B), Hes1 by immunostaining in LEC (added as Figure 5C) and both Hes1 and Hey1 by immunostaining in organ-cultured corneas (added as Figure 6C), in manuscript lines: 300-305, 311-320 and line 432-441 and as supplementary Figure S3, have been added to the manuscript. We are also in the process of looking into the other downstream targets of the Notch pathway, which is out of the scope of the present manuscript.
Point 2: In Figure 5 and 7, the western blots and the quantification needs to be elaborated as it is confusing. Preferably the controls should be before the treatment group. Try to set the control to 1 and find everything relative to that.
Response 2: Thanks for your suggestion. We did elaborate in Figure legend to make it clear. Since mimic and inhibitor have their own corresponding controls, each bar represents the changes of mimic or inhibitor compared to their corresponding controls shown as: M/MC and I/IC represent the ratio in comparison to one, to avoid too many bars graphs (Figure legends 5 and 6).

Reviewer 2 Report
The authors present here a follow up study on how the miR-146a regulates limbal epithelial cells homeostasis. The novelty of this manuscript resides on the fact that this microRNA specifically regulates Notch1, NUMB and Notch2 expression and has consequences in the stem cells maintenance of the limbal system. The manuscript is well written and the topic of how Notch regulates epithelial cells interesting for a broader audience, beyond the borders of the limbal system field.
Few elements require the author’s attention before publication:
- Can the authors comment on the fact that members of the Notch family (Hes, Hey, Dll/Jag ligands or N1-4 receptors) were not detected via the proteomics analysis (Fig 4)? Additionally, only Jag2 pops up in the transcriptomic, can the authors perform a GSEA analysis to confirm their enrichment results on Notch?
- Figure 6 would be improved by the addition of a staining for Numb. This result will also help the interpretation on the role of miR146a in stem cells renewal and asymmetric division (for example in case Numb is asymmetrically/differentially distributed).
- Staining for Notch1 should also be done in organ-cultured corneas to demonstrate its distribution in a 3D tissue
- Would usage of gamma-secretase inhibitors (e.g. DAPT) revert the effect on K15 cells in miR-146a transfected corneas? This would conclusively demonstrate that the effect of miR146a on stem cells depends on the activation of Notch.
Minor points:
In figure 5B the WB would benefit from a better loading control, as the first lane appears to contain a slightly lower amount of tot protein.
Supplementary data are not accessible
Author Response
Re: Manuscript ID, cells-886180, entitled " Integrated Transcriptome and Proteome Analyses Reveal the Regulatory Role of miR-146a in Human Limbal Epithelium via Notch Signaling". We want to thank the reviewers for a careful evaluation of our submitted paper. We have reviewed all of the concerns and have revised the paper accordingly.
Due to the reviewers’ comments and to make better structure and organization of manuscript, there are some changes in Figures. Figure 6 was changed as Figure 5C; Figure 7 was changed as Figure 6; Figure 8 was changed as Figure 7; a new figure was added as figure 8 (Effect of DAPT (Notch inhibitor) on miR-146aM transfected LECs by western blot). In addition, 3 figures were added as Supplementary Figures, S1, S2, S3. All the original western blots were submitted as Supplementary information.
A new author was added to the list of authors.
Response to Reviewer 2 Comments
Point 1: Can the authors comment on the fact that members of the Notch family (Hes, Hey, Dll/Jag ligands or N1-4 receptors) were not detected via the proteomics analysis (Fig 4)? Additionally, only Jag2 pops up in the transcriptomic, can the authors perform a GSEA analysis to confirm their enrichment results on Notch?
Response 1: This is a valuable suggestion. We performed a detailed gene set enrichment analysis (GSEA) and added pertinent information in section 3.1 and as Figure 1C and 1D, lines154-162, 231-238, 403-407. Some of the key genes as suggested by reviewer, like Notch1, DLL1, Hes, Jag, were found as enriched genes of Notch pathway (Figure 1D).
Point 2: Figure 6 would be improved by the addition of a staining for Numb. This result will also help the interpretation on the role of miR146a in stem cells renewal and asymmetric division (for example in case Numb is asymmetrically/differentially distributed).
Response 2: We agree with the reviewer; we included the staining for Numb, Notch2 and Hes1 as Figure 5C. Figure 6 was changed to Figure 5C. Also, we added this information in manuscript lines: 300-305.
Point 3: Staining for Notch1 should also be done in organ-cultured corneas to demonstrate its distribution in a 3D tissue
Response 3: We agree with the reviewer; we included the staining for Notch1 and also for Notch2, Numb, Hes1 and Hey1 in organ-cultured cornea as Figure 6C and supplementary Figure 3. Also, we added this information in manuscript lines: 312-320, 434-441.
Point 4: Would usage of gamma-secretase inhibitors (e.g. DAPT) revert the effect on K15 cells in miR-146a transfected corneas? This would conclusively demonstrate that the effect of miR146a on stem cells depends on the activation of Notch.
Response 4: This is an excellent point, indeed DAPT reverted the effect of K15 expression in miR-146aM transfected cells, which is added to Figure 8; also, in manuscript lines 357-364, 445-446.
Point 5: Minor points: In figure 5B the WB would benefit from a better loading control, as the first lane appears to contain a slightly lower amount of total protein. And supplementary data are not accessible.
Response 5: We agree with your point, however, the western blots were analyzed and normalized to the actin level using Odyssey CLX imaging system (LI-COR Biosciences). This method is quantitative and does not need ideal normalization by a housekeeping protein providing a relative expression.

Reviewer 3 Report
This is a nice study on the investigation of miR-146a targets in human primary limbal epithelial cells (LECs) using genomic and proteomic techniques. The study also confirmed the expression of some targeted genes/proteins in LECs and in organ-cultured corneas. The authors identified a number of targeted genes and pathways involved in stem cell maintenance and differentiation and found that miR-146a may have an opposite regulatory role in the fine-tuning of Notch-1 and Notch-2 expression in maintaining homeostasis of the corneal epithelial cells. I only have a few small comments to improve the manuscript.
1) Labels in Figure 1 A are too small;
2) Figure 7, I can not see bars in the bar graphs of HECE and Notch-2.
Author Response
Re: Manuscript ID, cells-886180, entitled " Integrated Transcriptome and Proteome Analyses Reveal the Regulatory Role of miR-146a in Human Limbal Epithelium via Notch Signaling". We want to thank the reviewers for a careful evaluation of our submitted paper. We have reviewed all of the concerns and have revised the paper accordingly.
Due to the reviewers’ comments and to make better structure and organization of manuscript, there are some changes in Figures. Figure 6 was changed as Figure 5C; Figure 7 was changed as Figure 6; Figure 8 was changed as Figure 7; a new figure was added as figure 8 (Effect of DAPT (Notch inhibitor) on miR-146aM transfected LECs by western blot). In addition, 3 figures were added as Supplementary Figures, S1, S2, S3. All the original western blots were submitted as Supplementary information.
A new author was added to the list of authors.
Response to Reviewer 3 Comments
Point 1. Labels in Figure 1 A are too small;
Response 1: Thank you, we changed the presentation as red and black clusters to make it clear in Figure 1A.
Point 2. Figure 7, I cannot see bars in the bar graphs of HECE and Notch-2.
Response 2. The bars are now included in all the Figures, Figure 7 was changed to Figure 6, which now included the bars.

Reviewer 4 Report
The authors have studied the Role of miR-146a in Human Limbal Epithelium via Notch Signaling through proteomic and transcriptomic approaches. Even though the study reports the regulation of Notch signaling by miR-146a for the first time in LECs, they never show the significance of this regulation.
(i) The authors have used mi-RNA mimic and mimic control, but have never shown the expression level of these RNA controls in cells after treatment. Even in the organ culture experiments the authors never show the level of this mi-RNA in control and other inhibitor treatments.
(ii) The authors have never justified why they focus on Notch when other signaling molecules are significantly changed in proteomic and transcriptomic profiling.
(iii) Also the authors did not show any downstream effect of this mi-RNA notch regulation. The study just describes indirect evidence for levels of notch in control and different conditions rather than showing any insight of this regulation.
(iv)
1. In figure 2, authors showed only 100 differentially expressed proteins in the heatmap. To get an overview, the authors need to provide the heatmap all the differentially expressed proteins (163 proteins).
2. Although the authors provided the supplementary table for the overlapping of the differentially expressed targets from the RNA-seq and proteomic data, for better understanding, the authors need to provide these details as a Venn diagram in the main figure.
3. The authors mentioned the pathway analysis in Figure 3, for choosing the notch pathway, have to talk about the changes in other molecules including mTOR, Ubiquitin ligase, etc, and tell why they were focussing only on Notch.
4. The Figure 4 shows the upstream regulators which are predicted by the IPA tool, it is always better show how many of these upstream regulators are differentially expressed in the dataset
5. In Figure 5, the control image is not clear. The authors need to provide a high resolution image for the control.
6. For Figure 8, the authors need to show the levels of mi-RNA under different treatment conditions.
Author Response
Re: Manuscript ID, cells-886180, entitled " Integrated Transcriptome and Proteome Analyses Reveal the Regulatory Role of miR-146a in Human Limbal Epithelium via Notch Signaling". We want to thank the reviewers for a careful evaluation of our submitted paper. We have reviewed all of the concerns and have revised the paper accordingly.
Due to the reviewers’ comments and to make better structure and organization of manuscript, there are some changes in Figures. Figure 6 was changed as Figure 5C; Figure 7 was changed as Figure 6; Figure 8 was changed as Figure 7; a new figure was added as figure 8 (Effect of DAPT (Notch inhibitor) on miR-146aM transfected LECs by western blot). In addition, 3 figures were added as Supplementary Figures, S1, S2, S3. All the original western blots were submitted as Supplementary information. A new author was added to the list of authors.
Response to Reviewer 4 Comments
Point 1: The authors have studied the Role of miR-146a in Human Limbal Epithelium via Notch Signaling through proteomic and transcriptomic approaches. Even though the study reports the regulation of Notch signaling by miR-146a for the first time in LECs, they never show the significance of this regulation.
(i) The authors have used mi-RNA mimic and mimic control but have never shown the expression level of these RNA controls in cells after treatment. Even in the organ culture experiments the authors never show the level of this mi-RNA in control and other inhibitor treatments.
Response 1: Thanks for your suggestions. Significant upregulation and downregulation of miR-146a expression levels were confirmed in mimic and inhibitor transfected cells by qRT-PCR compared to their corresponding controls and added as supplementary Figure S1 and mentioned in the manuscript lines 113-120, 211-212, 292-294.
Point 2: (ii) The authors have never justified why they focus on Notch when other signaling molecules are significantly changed in proteomic and transcriptomic profiling.
Response 2: Thank you for your suggestion. Notch is one of the significantly altered pathways in global transcriptomic and proteomic analysis (Figure 3 A-C). Moreover, the GSEA analysis (Figure 1 C and D) also showed a significant enrichment of Notch pathway. Therefore, it was logical to focus our further studies on Notch signaling pathway. We have added our justification for focusing on Notch signaling in the manuscript (line 65-70). However, we are also in the process of studying other significantly changed signaling pathways indicated in Figure 3A-C.
Point 3: (iii) Also, the authors did not show any downstream effect of this mi-RNA notch regulation. The study just describes indirect evidence for levels of notch in control and different conditions rather than showing any insight of this regulation.
Response 3: We appreciate pointing this out. The expression levels of Notch downstream targets, Hes1 by western blot (added to Figure 5B), Hes1 by immunostaining in LEC (added as Figure 5C) and both Hes1 and Hey1 by immunostaining in organ-cultured corneas (added as Figure 6C), in manuscript lines: 300-305, 311-320 and line 432-441 and as supplementary Figure S3, have been added to the manuscript. We are also in the process of looking into the other downstream targets of the Notch pathway, which is out of the scope of the present manuscript.
Point 4: In figure 2, authors showed only 100 differentially expressed proteins in the heatmap. To get an overview, the authors need to provide the heatmap all the differentially expressed proteins (163 proteins).
Response 4: We appreciate your comment; we did change the heatmap for all the differentially expressed proteins (163 proteins) in Fig 2B.
Point 5. Although the authors provided the supplementary table for the overlapping of the differentially expressed targets from the RNA-seq and proteomic data, for better understanding, the authors need to provide these details as a Venn diagram in the main figure.
Response 5: We do agree with reviewer. We have provided a Venn diagram for the differentially expressed targets from the RNA-seq and proteomic data as Supplementary Figure S2.
Point 6: The authors mentioned the pathway analysis in Figure 3, for choosing the notch pathway, have to talk about the changes in other molecules including mTOR, Ubiquitin ligase, etc, and tell why they were focusing only on Notch.
Response 6: Thank you for your comment. We added our justification in the manuscript, line 65-70, 267-270, why we focused on Notch signaling. We added a couple of sentences about changes in other molecules (line 259-267). However, we are in the process of looking into other significantly altered signaling molecules as well.
Point 7: The Figure 4 shows the upstream regulators which are predicted by the IPA tool, it is always better show how many of these upstream regulators are differentially expressed in the dataset
Response 7: Thank you for your suggestion, the significant upstream regulators are labeled in red in the Figure 4.
Point 8: In Figure 5, the control image is not clear. The authors need to provide a high-resolution image for the control.
Response 8: We provided high resolution of the image, Figure 5.
Point 9: For Figure 8, the authors need to show the levels of mi-RNA under different treatment conditions.]
Response 9: Thanks for your suggestion, significant upregulation and downregulation of miR-146a expression levels were confirmed in mimic and inhibitor transfected cells by qRT-PCR compared to their corresponding controls and added as supplementary Figure S1 and mentioned in the manuscript lines 113-120, 211-212, 292-294.

Round 2
Reviewer 1 Report
The changes made to the manuscript by the authors are sufficient to answer the concerns.
Author Response
Response to Reviewer 1 Comments
Re: Manuscript ID, cells-886180, entitled " Integrated Transcriptome and Proteome Analyses
Reveal the Regulatory Role of miR-146a in Human Limbal Epithelium via Notch Signaling". We
want to thank the reviewers for a careful evaluation of our submitted paper.
Reviewer 2 Report
The authors answered all my concerns in a satisfactory manner
Author Response
Response to Reviewer 2 Comments
Re: Manuscript ID, cells-886180, entitled " Integrated Transcriptome and Proteome Analyses
Reveal the Regulatory Role of miR-146a in Human Limbal Epithelium via Notch Signaling". We
want to thank the reviewers for a careful evaluation of our submitted paper.
Reviewer 4 Report
Hi,
The authors have done revisions to some extent according to the suggestion.
(1) The first concern that authors need to justify why they choose Notch signaling needs further justification apart from the factor that it is commonly affected in both transcriptomic and proteomic data. Based on the list provided it is not one of the top differentially expressed genes in both data sets. Authors need to show why they chose Notch over other differentially expressed genes.
(2)The authors did not add any references in the introduction to support the role of Notch in regulating homeostasis and wound healing.
(3)In Fig.5C Hes1 do not show a significant difference between M and MC. Authors need to explain. Mention the n number for the IF images.
Author Response
Response to Reviewer 4 Comments
Re: Manuscript ID, cells-886180, entitled " Integrated Transcriptome and Proteome Analyses Reveal the Regulatory Role of miR-146a in Human Limbal Epithelium via Notch Signaling". We want to thank the reviewers for a careful evaluation of our submitted paper. We have reviewed all of the concerns and have revised the paper accordingly.
To answer to the reviewers’ comments and to make better structure and organization of the manuscript, we made some changes in the Figures. A new Supplementary figure was added as figure S3 (Identification of pathways that are most significantly altered in RNA-seq in miR-146aM compared to mimic control transfected LECs.) Previous Supplementary Figure S3 is now S4.
The authors have made text revisions according to the suggestions (all the changes in purple color).
Point 1: The first concern that authors need to justify why they choose Notch signaling needs further justification apart from the factor that it is commonly affected in both transcriptomic and proteomic data. Based on the list provided it is not one of the top differentially expressed genes in both data sets. Authors need to show why they chose Notch over other differentially expressed genes.
Response 1: Thank you for your suggestion. As it is mentioned in the manuscript, Notch is one of the most significantly altered pathways in global transcriptomic and proteomic analysis (Figure 3 A-C). Moreover, the GSEA analysis (Figure 1 C and D) also showed a significant enrichment of Notch pathway.
Further, in order to define the relative significance of differentially altered pathways in our transcriptomic data, we followed the method described by Chen et al 2013 [Chen, E.Y., Tan, C.M., Kou, Y. et al. Enrichr: interactive and collaborative HTML5 gene list enrichment analysis tool. BMC Bioinformatics 14, 128 (2013)] to rank the significantly enriched pathways (FDRp<0.05) based on a combined enrichment score and odd ratio. Briefly, the combined score can be described as c = log(p) * z, where c = the combined score, p = Fisher exact test p-value, and z = z-score for deviation from expected rank. Odd ratio defines odds of an event happening as the likelihood that an event will occur, expressed as a proportion of the likelihood that the event will not occur. An OR >1 indicates increased occurrence of event OR <1 indicates decreased occurrence of event.
We have added these results in Supplementary Figure S3. Notch signaling as one of the most significantly altered pathways (FDRp <0.05) with a maximum combined enrichment score (85) and Odd ratio (12). Therefore, based on these results as well keeping in view the importance of Notch signaling in homeostasis and wound healing, we further studied the Notch signaling. We also mentioned before that Notch signaling is a distinct pathway of interest in limbal epithelial stem cell maintenance, which is known to regulate corneal epithelial homeostasis and controls cell-fate specification events during wound healing.
We added more justification for focusing on Notch signaling in the manuscript (line 68-69, 155-158, 269-271) and supplementary Figure S3.
Point 2: The authors did not add any references in the introduction to support the role of Notch in regulating homeostasis and wound healing.
Response 2: We appreciate pointing this out. References were added (line 71).
Point 3: In Fig.5C Hes1 do not show a significant difference between M and MC. Authors need to explain. Mention the n number for the IF images.
Response 3: We appreciate your comment; n number was added in Figure 5C and 6C legends and explained in the manuscript. We agree with the reviewer that Hes1 expression level did not show significant changes in miR-146aM transfected cultured cells by immunostaining and western blot. A possible explanation is provided in the discussion section (line 467-479).
Published data convincingly showed that Hes1 is activated in other systems by Notch-2 [Hayashi et. al., 2015] and probably to some extent by Notch-1. In the cornea, miR-146aM upregulated Notch-1 but also significantly downregulated Notch-2. Thus, the net effect on Hes1 upon miR-146aM transfection may be minimal due to its opposing action on Notch-1 and Notch-2 in limbal epithelium. In addition, other downstream targets of Notch signaling such as HeyL, Hes6, P21 (CDKN1A), and/or Cyclin D3 (CCND3) may be involved in the activation of Notch signaling in limbal epithelial homeostasis, which needs further investigation.
